# Fluidized ZnO@BCFPs Particle Electrodes for Efficient Degradation and Detoxification of Metronidazole in 3D Electro-Peroxone Process

**DOI:** 10.3390/ma15103731

**Published:** 2022-05-23

**Authors:** Dan Yuan, Shungang Wan, Rurong Liu, Mengmeng Wang, Lei Sun

**Affiliations:** 1School of Chemical Engineering and Technology, Hainan University, Haikou 570228, China; 21110817000020@hainanu.edu.cn (D.Y.); 993994@hainanu.edu.cn (S.W.); 20085600210112@hainanu.edu.cn (M.W.); 2Key Laboratory of Advanced Materials of Tropical Island Resources, Ministry of Education, Haikou 570228, China; 3Key Laboratory of Solid Waste Resource Utilization and Environmental Protection of Haikou City, Haikou 570228, China; 4School of Materials Science and Engineering, Hainan University, Haikou 570228, China; 20180411310048@hainanu.edu.cn

**Keywords:** electro-peroxone process, particle electrode, metronidazole, degradation mechanism, detoxification

## Abstract

A novel material of self-shaped ZnO-embedded biomass carbon foam pellets (ZnO@BCFPs) was successfully synthesized and used as fluidized particle electrodes in three-dimensional (3D) electro-peroxone systems for metronidazole degradation. Compared with 3D and 2D + O_3_ systems, the energy consumption was greatly reduced and the removal efficiencies of metronidazole were improved in the 3D + O_3_ system. The degradation rate constants increased from 0.0369 min^−1^ and 0.0337 min^−1^ to 0.0553 min^−1^, respectively. The removal efficiencies of metronidazole and total organic carbon reached 100% and 50.5% within 60 min under optimal conditions. It indicated that adding ZnO@BCFPs particle electrodes was beneficial to simultaneous adsorption and degradation of metronidazole due to improving mass transfer of metronidazole and forming numerous tiny electrolytic cells. In addition, the process of metronidazole degradation in 3D electro-peroxone systems involved hydroxyethyl cleavage, hydroxylation, nitro-reduction, N-denitrification and ring-opening. The active species of ·OH and ·O_2_^−^ played an important role. Furthermore, the acute toxicity LD_50_ and the bioconcentration factor of intermediate products decreased with the increasing reaction time.

## 1. Introduction

Antibiotics are one of the primary antibiotic groups that are widely used to treat various bacterial infections or inhibit pathogenic microorganisms, and they are even used as growth promoters in animal husbandry and agriculture [1]. Considering the limited removal capacity of traditional wastewater treatment plants for antibiotic pollutants [2,3], they have been detected in the aquatic environment such as lakes [4], rivers [5,6], and groundwater [7]. Antibiotics may exist for a long term in the natural environment and cause the generation of antibiotic-resistant pathogenic bacteria and antibiotic-resistant genes [8]. Therefore, antibiotics in wastewaters need to be adequately removed before their discharge into the water environment.

Various techniques have been studied to treat antibiotics in aquatic environments, including adsorption [9,10], photocatalysis [11,12], electrocatalysis [13,14], flocculation and coagulation [15]. Among them, the photocatalytic advanced oxidation technology has received increasing attention due to its strong oxidation capability, which can completely mineralize antibiotics to CO_2_ and H_2_O by oxidants and active species generated through a series of photogenerated electrons and holes reaction on the photocatalysts [16,17]. However, the photocatalytic efficiency is limited by the adsorption capacity of photocatalysts and photogenerated electron-hole recombination. Compared with photocatalysis, electrochemical advanced oxidation processes (EAOPs) could directly in-situ generate active species on the surface of electrodes for the degradation of refractory organic pollutants [18]. Among EAOPs, three-dimensional (3D) electrocatalytic technology refers to the loading of some granular materials with catalytic activity as particle electrodes into two-dimensional (2D) electrode reactors, which enhances the mass transfer of pollutants by adsorption and promotes pollutant decomposition by forming numerous tiny electrolytic cells [19]. In addition, O_3_ is a selective oxidant with an oxidation potential of 2.07 V, and it can rapidly oxidize active double bond organic compounds such as alkenes, amines, and reductive sulfides [20]. Further enhancing refractory organic pollutant removal and shortening the reaction time are effective methods when combined with 3D electrocatalytic system and ozone catalytic oxidation during the treatment process to form a 3D electro-peroxone system. Thus, the active specie ·OH production could be strengthened by the reaction between O_3_ and H_2_O_2_ that is in situ electro-generated on the cathode (Equations (1) and (2)), and it can contribute to oxidize organic pollutants without selectivity [21].
(1)$$O_{2}+2H^{+}+2e^{-}\to H_{2}O_{2}$$
(2)$${2O}_{3}+H_{2}O_{2}\to 2\cdot OH+3O_{2}.$$

The 3D electro-peroxone system has been explored to treat various wastewaters, and the synergistic degradation of pollutants in pharmaceutical wastewater can be achieved by the combination of 3D electrochemical process and ozonation [22,23]. Compared with individual ozonation and the 3D electrocatalytic system using granular activated carbon (GAC) as the particle electrodes, the 3D electro-peroxone system considerably enhanced total organic carbon (TOC) abatement to ∼71% and reduced the inhibition of the luminescent bacterial to <70% in microtox bioassays [24]. Notably, particle electrodes also play an important role in the 3D electro-peroxone system because they were polarized in the electric field to form a large number of bipolar microelectrodes, which greatly increased the electrochemical reaction center to strengthen the pollutant removal. Thus, the selection of the particle electrode is the key point of the 3D electro-catalytic oxidation system. At present, the commonly used 3D particle electrodes mainly include conductive particles of low and high impedance in accordance with their properties, such as GAC [25], Al_2_O_3_ [26], and Kaolin [27] carriers to prepare a composite particle electrode catalyst. However, typical carriers of the 3D particle electrode with high density leads to the stack of particle electrodes in the system. This phenomenon easily produces a short circuit current and reduces the efficiency of electrocatalysis. Therefore, preparing a lightweight shaped particle electrode with high catalytic activity to avoid the stack of particle electrodes is difficult.

In this work, a novel material of self-shaped ZnO-embedded biomass carbon foam pellets (ZnO@BCFPs) was prepared as ideal 3D particle electrodes to construct a 3D electro-peroxone system. Metronidazole as a typical nitroimidazole antibiotic, which has high solubility, low biological degradation, and high carcinogenic and mutagenic properties, was selected as the representative target pollutant. This study aimed to investigate the synergy effect of 3D electro-peroxone in terms of metronidazole removal compared with the single treatment process and evaluate the relative contribution of reactive oxidation species. The evolution of metronidazole and its toxicity was explored as well.

## 2. Materials and Methods

### 2.1. Materials and Chemicals

Eucalyptus sawdust of 100 mesh was derived from timber processing plants in Liuzhou, Guangxi Province, China. The composition of the major components of the eucalyptus powder was 41% cellulose, 31% hemicellulose and 29% acid-insoluble lignin [28]. Metronidazole (purity 99%, C_6_H_9_N_3_O_3_) was purchased from Aladdin Industrial Corporation, Shanghai, China. Phenol, H_2_SO_4_ (98%), methanol, NaOH, formaldehyde (37%), absolute ethanol, ZnCl_2_, HCl (37%), benzoquinone, and isopropanol were of analytical grade and used directly without further purification.

### 2.2. Preparation of ZnO@BCFPs Particle Electrodes

Lightweight shaped carbon foam pellets derived from eucalyptus sawdust were prepared successfully following our previous study [9,29]. Briefly, 20 g of eucalyptus sawdust, 60 g of phenol, and 1.96 mL of H_2_SO_4_ were mixed in a 250 mL three-necked flask, and liquefaction reaction was initiated at 150 °C under vigorous stirring for 2 h. After cooling, liquefied product was washed with methanol and filtrated. Then, filtrate was placed into a rotary evaporator for vacuum distillation at 50 °C to remove methanol. Half of the obtained liquefied product, 1.44 g of NaOH, and 60 mL of deionized water were mixed in a 500 mL three-necked flask at 70 °C and stirred for 5 min, then 34 mL formaldehyde was added dropwise and stirred for 30 min. Thereafter, 60 mL of absolute ethanol and 20 mL of water were supplied to the mixture solution and stirred continuously for another 2 h. Finally, the mixture was diluted with 280 mL of water and was then transferred into Teflon reactor and underwent hydrothermal reaction at 130 °C for 24 h. The obtained hydrothermal product was thoroughly mixed with ZnCl_2_ and then filled into spherical molds. The mass ratios of ZnCl_2_ to eucalyptus sawdust were 2.4, 3.2, 4.0 and 4.8, respectively. After drying at 85 °C, the carbonaceous pellets were calcined in a tubular furnace at 600 °C for 90 min under nitrogen atmosphere to obtain the lightweight ZnO@BCFPs.

### 2.3. Characterization

The surface morphology of ZnO@BCFPs was observed with scanning electron microscopy (SEM, Thermoscientific Verios G4 UC, Waltham, MA, USA). The crystal structure of the ZnO@BCFPs particle electrode was analyzed by X-ray powder diffraction analysis (XRD, Bruker D8 Advance, Karlsruhe, Germany) with Cu-K_α_ radiation source (λ = 1.5418 Å). The functional groups and chemical bonds on the surface of particle electrode were obtained with a Fourier transform infrared spectrometer (FTIR, Bruker T27, Karlsruhe, Germany). A high-resolution transmission electron microscope (HRTEM, Thermoscientific Talos F200X G2, Waltham, MA, USA) was used to analyze lattice fringes in the crystal structure of materials. X-ray photoelectron spectroscopy (XPS, Thermoscientific Escalab 250Xi, Waltham, MA, USA) of ZnO@BCFPs was obtained to determine element composition, chemical states and electronic states. Electrochemical impedance spectroscopy (EIS) was measured by a CHI660E electrochemical work station (Chenhua Instrument, Shanghai, China) with a standard three electrode system (Pt counter electrode, Ag/AgCl reference electrode), and the frequency was in the range of 0.1–10^6^ Hz. The EIS measure was operated in 0.05 mol L^−1^ Na_2_SO_4_ solution. The working electrode was prepared as follows: 20 mg sample was dispersed in a mixture of 20 μL Nafion solution (5%, DuPont) and 1 mL absolute ethanol and then ultrasonically processed for 10 min. After that, 20 μL of the slurry was dipped onto the surface of the 1 cm × 1 cm indium tin oxide glass, and then dried in an oven at 80 °C.

### 2.4. Experimental Setup of 3D Reactor

Electrolysis experiments were conducted in a self-made 3D electro-peroxone system, as shown in Figure 1. A single chamber electro-peroxone system with an effective size of 120 mm × 80 mm × 130 mm was constructed using acrylic plexiglass. The ruthenium iridium-coated titanium (RuO_2_-IrO_2_/Ti) plate and stainless-steel plate with the same size of 50 mm × 75 mm were used as anode and cathode, respectively. They were connected by DC power supply (KPS-3005D, ZHAOXIN^®^, Shenzhen, China). The lightweight ZnO@BCFPs particle electrodes were filled between the two porous electrode baffles to avoid the occurrence of short-circuit current. At the bottom of the 3D electro-peroxone system, a demountable baffle with uniform aperture distribution was also installed. An aerator was arranged under the baffle and connected with a silent double-hole aeration pump (SB-748, Zhongshan Songbao Electric Appliance Co., Ltd., Zhongshan, China) or ozone generator (FL-803AS, Shenzhen Feili Electrical Technology Co., Ltd., Shenzhen, China). Air as a gas source entered the reactor directly through the aeration pump, or it was converted to O_3_ by the ozone generator and then into the reactor. The airflow of the aeration could blow up the lightweight particle electrode and then fall down, which formed the inner circulation flow of the particle electrode in the main reaction chamber. At the same time, the liquid in the main reaction chamber rose and entered the cathode and anode chamber, respectively. Then, it entered the main reaction chamber from the bottom of the baffle, which formed the liquid inner circulation and the fluidized particle electrode. Therefore, the particle electrode could contact the organic compound adequately.

### 2.5. Electrocatalytic Experiments

All electrocatalytic experiments were conducted under galvanostatic conditions. In the electrocatalytic experiment, a 600 mL of 15 mg L^−1^ metronidazole aqueous solution was used as simulated wastewater, and the distance between anode and cathode plate was adjusted to 4 cm. The effect of the aeration rate, electrolyte Na_2_SO_4_ concentration, current density, and particle electrode dosage and solution pH in the main reaction chamber were investigated by adjusting the range of parameters. A total of 1.5 mL of sample was collected at times of 0, 5, 10, 20, 40, 60, 80, 100, and 120 min, and the samples were filtered through a 0.25 μm filter to analyze the concentration of metronidazole. The metronidazole concentration was determined by a high-performance liquid chromatography (HPLC, Agilent 1260, CA, USA) equipped with a ZORBAX Eclipse Plus C18 column (250 mm × 4.6 mm, 5 μm) at the maximum absorption wavelength of 320 nm. Ultrapure water and HPLC-methanol (80:20 *v*/*v*) were used as the mobile phase at a flow rate of 1 mL min^−1^. The calculation formula for removal efficiency is shown as follows:(3)$$RE=\frac{C_{0}-C_{t}}{C_{0}}=1-\frac{C_{t}}{C_{0}},$$
where *C*_0_ and *C_t_* (mg L^−1^) represent the concentration of metronidazole at initial and time *t* (min), respectively. The electrocatalytic degradation intermediate products of metronidazole were analyzed using Shimadzu LCMS-IT-TOF equipped with a C18 column. The mineralization degree of metronidazole under optimal degradation conditions was measured by TOC analyzer (Shimadzu TOC-L, Kyoto, Japan).

## 3. Results and Discussion

### 3.1. Characterization

Figure 2a shows the XRD patterns of the particle electrode prepared with the mass ratio of ZnCl_2_ to eucalyptus sawdust of 3.2:1. The particle electrode had many obvious diffraction peaks, which were indicative of a good crystallinity. The crystal of ZnO with a hexagonal structure was confirmed by comparing with PDF cards (PDF#80-0075) [30], which was consistent with the XRD results of ZnO prepared by Nie et al. [31]. ZnO is an eco-friendly catalyst, which has the environmental superiority of heterogeneous catalysis that can meet the need of organic pollutant degradation. As shown in Figure 2b, the band at 3385 cm^−1^ was attributed to the stretching vibration of O–H bond from BCFPs. The band at approximately 744 cm^−1^ was assigned to the angular deformation of hydroxyl group [32]. The band at 601 cm^−1^ could be attributed to the vibration characteristic of the Zn–O bond [33]. The band range of 1430–1611 cm^−1^ corresponded to C=C stretching vibration of the aromatic ring skeleton in BCFPs [34,35]. The band at 1611 cm^−1^ may also be attributed to the C=O stretching vibration of carbonyl groups in BCFPs [36]. The bands around 1993 cm^−1^ and 2116 cm^−1^ were ascribed to C=O and O-H stretching vibration, respectively [37].

The interface conductivity of ZnO@BCFPs with different mass ratios of ZnCl_2_ to eucalyptus sawdust was illustrated by EIS. The Nyquist plots with fitted data and an equivalent circuit were presented in Figure 2c, and the fitted parameters were listed in Table 1. In equivalent circuits, R_0_, R_1_ and R_2_ represent ohmic resistance, charge transfer resistance, and diffusion resistance of particle electrode, respectively; C_1_ and C_2_ represent the double-layer capacities associated with charge and mass diffusion [38]. The particle electrode prepared with the mass ratio of ZnCl_2_ to eucalyptus sawdust of 3.2:1 with a smaller R_1_ value (45.40 Ω) facilitated the transfer of electrons than the other particle electrodes.

The results of XPS characterization were shown in Figure 3. From the total survey (Figure 3a), it can be observed that the particle electrode mainly contains C, O and Zn elements. In the C 1s spectrum (Figure 3b), the peaks at 284.3 eV, 286.2 eV and 288.4 eV were charged to C=C, C-O and C=O [39]. In the O 1s spectrum (Figure 3c), the peaks at 530.2 eV and 531.1 eV were attributed to Zn-O [40,41]. The peaks at 531.7 and 532.8 eV were corresponded to carbonyl oxygen atoms or oxygen atom in hydroxyl groups [42], and 533.7 eV was assigned to O=C-O [39]. The banding energies at 1022.0 eV and 1045.0 eV in the Zn 2p spectrum were attributed to Zn 2p_3/2_ and Zn 2p_1/2_, respectively, which were assigned to the lattice zinc in ZnO (Figure 3d).

As shown in Figure 4a, the prepared particle electrode was spherical with a particle size of about 5 mm. SEM was used to further observe the micro-morphology of the particle electrode. A large number of rectangular-, pentagonal-, polygonal-, and maple-shaped layered catalysts were observed on the smooth external and internal surfaces of the particle electrode; they had a size of around 1–3 μm (Figure 4b,c).

The microstructural feature of the particle electrode was further analyzed by HRTEM, as presented in Figure 5. It was clearly exhibited that the lattice fringes of ZnO were 0.25 nm and 0.28 nm, corresponding to crystal planes of (101) and (100) (Figure 5a,b), respectively, which agreed with the results given by XRD characterization. In addition, the amorphous region of BCFPs was observed. The crystal lattice of ZnO was interlaced with the amorphous region of BCFPs (Figure 5c), indicating that ZnO was evenly distributed on the surface of BCFPs, which was consistent with SEM results.

### 3.2. Effects of ZnCl_2_ Dosage on Removal Efficiency

ZnCl_2_ dosage determines the amount of catalyst loaded on the particle electrode. Thus, it greatly influences the removal of metronidazole in wastewater. As shown in Figure 6, the removal efficiencies of metronidazole over time were nearly linear in the 2D electrolysis system, and the removal efficiency was 65.5% at 120 min. The decomposition of metronidazole only happened on the surface of cathode and anode under fixed experimental conditions, so the removal efficiencies of metronidazole nearly increased linearly with increasing reaction time. However, after adding the ZnO@BCFPs particle electrode into 2D electrocatalytic reactor to form 3D electrocatalytic system, the removal efficiencies of metronidazole increased non-linearly with increasing reaction time due to the adsorption and catalytic degradation simultaneously occurred on the surface of ZnO@BCFPs. At the initial stage of electrocatalysis, the rapid removal rate of metronidazole was attributed to a large number of adsorption sites on the surface of the particle electrode, and the adsorption rate was greater than the degradation rate. As the reaction progressed, the adsorption sites on the particle electrode surface were gradually occupied by metronidazole molecules and intermediate products, resulting in a slow enhancement in the removal rate. Therefore, the removal efficiencies of metronidazole over time were non-linear.

The removal efficiencies were increased by 28.1%–32.5% after the addition of particle electrodes, which indicated that the 3D electrocatalytic system constructed by particle electrodes can promote the abatement of metronidazole. When the mass ratio of ZnCl_2_ to eucalyptus sawdust increased from 2.4:1 to 3.2:1, the electrocatalytic activity of the particle electrode was significantly improved, and the removal efficiency was up to 98.0% after 120 min. This enhancement was attributed to the synergistic effect of adsorption, electrosorption, and electrocatalytic oxidation of the particle electrodes in the 3D electrocatalytic system. Notably, adsorption was responsible for accelerating the degradation rate by increasing the concentration of contaminants at the interface of the particle electrodes [43]. However, when the mass ratio was further increased to 4.0:1–4.8:1, the total removal efficiency became worse, which may be related to their adsorption properties. Figure 6 clearly indicates that the adsorption capacity of the particle electrode was positively correlated with the electrocatalytic degradation capacity. Therefore, the particle electrode prepared with the mass ratio of ZnCl_2_/eucalyptus sawdust of 3.2:1 was selected to study the influence of key kinetic factors on metronidazole removal efficiency.

### 3.3. Effects of Kinetics Parameters on Removal Efficiency

The influences of electrolyte concentration, current density, particle electrode dosage, and initial solution pH on metronidazole removal efficiency were investigated, as presented in Figure 7. The Na_2_SO_4_ electrolyte used in the electrocatalytic system has two functions which can synergistically degrade contaminants [44]: one is to reduce the ohmic resistance for improving the conductivity of solution, which can accelerate the electron transfer and the degradation rate of pollutants; the other is to produce stronger active species, such as S_2_O_8_^2−^ (*E*^0^ = 2.1 V) and SO_4_^•−^ (*E*^0^ = 2.6–3.2 V), which may be produced when using Na_2_SO_4_ as an electrolyte. As presented in Figure 7a, the removal efficiency of metronidazole was found to increase from 70.5% to 83.4% within 60 min when the concentration of Na_2_SO_4_ increased from 0.025 mol L^−1^ to 0.05 mol L^−1^. However, the removal efficiency decreased when the concentration further increased to 0.1 mol L^−1^. This result was due to that the degradation may be adversely affected by the adsorption of active substances and the reduction in active sites on the particle electrode surface [45]. Moreover, the high electrolyte concentration with high electrical conductivity resulted in increased bypass current and decreased reaction current in the 3D electrocatalytic system [46]. In addition, the excessive concentration will cause the waste of electrolyte and increase the difficulty of subsequent wastewater treatment [19]. Therefore, the optimal Na_2_SO_4_ concentration was 0.05 mol L^−1^ in the 3D electrocatalytic system.

As shown in Figure 7b, the removal efficiency within 60 min increased from 72.5% to 83.4% when the current density increased from 5.33 mA cm^−2^ to 8.00 mA cm^−2^. This is because the current density affects the particle electrode repolarization, which refers to inducing the filled particles with high impedance to form positive and negative poles at both ends by the application of an external voltage to the main electrode. Thus, the repolarization process can form numerous tiny electrolytic cells, and further contribute to the electrochemical oxidation efficiency. Meanwhile, the amount of H_2_O_2_ produced by the cathode also increased due to the increase in mass transfer rate between electrons [47], which resulted in a large increase in ·OH and the regeneration of catalyst supported on the carbon-based particle electrode. Ultimately, the removal efficiency of metronidazole improved. However, the removal efficiency dropped to 66.2% when the current density further increased to 10.67 mA cm^−2^. This result may be due to the fact that the excessive current density aggravated the occurrence of side reactions, such as oxygen evolution at anode and hydrogen evolution at cathode and heat generation, which resulted in a reduction in the production of ·OH, as well as increased bypass current; thus, it was not conducive to the degradation reaction [19,46]. Excessive electrolytic current density with low utilization of electrical energy will also increase energy consumption, which was unfavorable to the practical application of the 3D electrocatalytic system.

The effect of particle electrode dosage on the 3D electrocatalytic system is shown in Figure 7c. The removal efficiencies of metronidazole were 65.3% and 74.4% for dosages of 6.67 g L^−1^ and 10.00 g L^−1^ within 40 min, respectively. This result was due to the incremental number of particle electrodes involved in the induced charge increasing the number of micro-electrolytic cells and then the total active sites and reaction area, which can produce more active species such as ·OH. Ultimately, the removal efficiency of metronidazole enhanced. Moreover, the addition of particle electrodes improved the electrolytic efficiency, which reduced the electric energy [43]. However, the removal efficiency only increased slightly to 75.2% within 40 min when the dosage increased to 13.33 g L^−1^. Considering that the excessive amount of catalyst will increase the cost and limit the practical application, 10.00 g L^−1^ was selected as the optimal particle electrode dosage.

According to a previous study [48], solution pH will affect the surface adsorption performance of particle electrodes and the generation of H_2_O_2_, ·OH, and other active oxygen species in the electrocatalytic system, which affected its electrocatalytic degradation performance. As presented in Figure 7d, the acidic environment of the system was unfavorable to the catalyst at pH = 5. Given that ZnO is very sensitive to acidic pH, acid will destroy the layer structure of ZnO and reduce its catalytic ozonation activity. Liu et al. [49] also reported that an obvious leaching of zine ions will occur when solution pH < 5.8. When the system was alkaline, the removal ability of metronidazole was decreased because of the reduction in the oxidation ability of hydroxyl group and the decomposition of H_2_O_2_ into H_2_O and O_2_. The catalyst ZnO may be decomposed into soluble zine salt under alkaline pH condition, which resulted in the decline in catalytic performance. Meanwhile, hydroxyl radical scavengers such as carbonate and bicarbonate may be generated in alkaline solution [50]. The removal efficiency of metronidazole is the best when the solution pH is neutral due to the synergistic effect of adsorption and electrocatalysis of the particle electrode.

### 3.4. Comparison of Removal Efficiency in Different Systems

As manifested in Figure 8a,b, the degradation kinetics of metronidazole in different systems including 2D electrolysis (2D), 2D electrolysis + O_3_ (2D + O_3_), 3D electrocatalysis (3D), 3D electrocatalysis + O_3_ (3D + O_3_), and ZnO@BCFPs + O_3_ (no electricity) were studied by the pseudo-first-order kinetic model (Equation (4)), and the removal performance was compared clearly.
(4)$$\ln(\frac{C_{0}}{C_{t}})=-k_{\mathrm{obs}}t,$$
where *k*_obs_ is the apparent rate constant (min^−1^). The fitting results and calculated *k*_obs_ were presented in Table 2.

In the 2D system, the removal efficiency of metronidazole was 65.5% after 120 min at an air aeration rate of 0.4 L min^−1^. The removal efficiency in the 3D system was up to 100% after the particle electrode was added. The value of *k*_obs_ was 4.17 times higher in the 3D system (0.0369 min^−1^) than that in the 2D system (0.0089 min^−1^). The results showed that the addition of particle electrodes greatly enhanced the abatement of metronidazole, which was mainly due to the increment in adsorption and catalytic active sites and the acceleration of mass transfer. This condition was more advantageous to the in-situ degradation of metronidazole on its surface. The removal efficiency in 2D + O_3_ system was enhanced greatly compared with that in the 2D system, and *k*_obs_ was increased by 3.79 times. Compared with the 3D system, the enhancement effect of degradation was also found in the 3D + O_3_ system, which is popularly known as the electro-peroxone process. The complete degradation of metronidazole took 120 min in the 3D system, but the degradation time was shortened to 60 min in the 3D + O_3_ system, and the value of *k*_obs_ increased to 1.50 times. In the ZnO@BCFPs + O_3_ system, the removal efficiency of metronidazole even reached 100% within 80 min without the electric field. All the above-mentioned results indicated that O_3_ played a significant role in metronidazole degradation. This finding was mainly ascribed with two factors: (i) O_3_ itself had the ability to oxidize organic substances and could react with H_2_O_2_ produced by the cathode to generate ·OH with higher oxidation potential; (ii) catalyst ZnO could effectively promote the decomposition of O_3_ to produce reactive oxygen species such as ·OH and ·O^2−^ [49].

The energy consumption is usually evaluated by the electric energy per mass (*E*_EM_) and the electric energy per order (*E*_EO_) in the electrochemical degradation process. *E*_EM_ is defined as the energy consumption per unit mass of metronidazole degraded, and it was estimated from Equation (5) [51]. *E*_EO_ is the electric energy in kilowatt hours required to degrade the contaminant by one order of magnitude (90%) in a unit volume of polluted water in batch operation mode, and the calculation formula for *E*_EO_ is shown as follows [51,52]:(5)$$E_{EM}=\frac{E_{\mathrm{cell}}It}{V\left(C_{0}-C_{t}\right)}$$
(6)$$E_{\mathrm{EO}}{(\mathrm{kWh}\;m}^{-3}{\;\mathrm{order}}^{-1})=\frac{E_{\mathrm{cell}}It}{V\log(C_{0}/C_{t})},$$
where *E*_cell_ is the average cell potential (V), *I* is the applied current (A), *t* is electrolysis or electrocatalysis time (h), *V* is the solution volume (L). Owing to log(*C*_0_/*C*_t_) = 0.4343·*k*_obs_*t*, the *E*_EO_ expression can be simplified to Equation (7) assuming first-order kinetics.
(7)$$E_{\mathrm{EO}}{(\mathrm{kWh}\;m}^{-3}{\;\mathrm{order}}^{-1})=\frac{0.038E_{\mathrm{cell}}I}{Vk_{\mathrm{obs}}}.$$

The values of *E*_EM_ and *E*_EO_ calculated in different systems are listed in Table 2. Obviously, the values of *E*_EM_ and *E*_EO_ in the 2D system were both higher than those in the 3D system. Specifically, the *E*_EM_ and *E*_EO_ values in the 3D + O_3_ system were 188.21 kWh kg^−1^ and 2.57 kWh m^−3^, respectively, which were significantly lower than those in the 2D + O_3_ system under the same conditions. The results showed that the introduction of particle electrodes into the electrocatalytic system reduced the energy consumption.

The aeration rate of O_3_ is the key factor affecting the degradation rate of metronidazole by catalytic ozonation. As shown in Figure 8c, the time for metronidazole to be totally degraded shrank from 100 min to 60 min when O_3_ aeration rate increased from 0.2 L min^−1^ to 0.4 L min^−1^. This significant enhancement in degradation performance was due to more reactive oxygen species such as ·OH and ·O_2_^−^ being generated with increasing O_3_ concentration in solution. However, a further increase in the O_3_ aeration rate reduced the enhancement in the degradation effect. This result may be due to the excess O_3_ reducing the conductivity of solution [47] or bubbles generated from excessive aeration leading to insufficient contact between the solution and the anode and/or particle electrode, which slowed down the process of the reaction. The experimental data were fitted by pseudo-first-order dynamics, and the parameter results are shown in Table 2. Each batch of experiments exhibited a good linear fit (Figure 8d). With the increase in O_3_ aeration rate (from 0.2 L min^−1^ to 0.6 L min^−1^), *k*_obs_ increased from 0.0358 min^−1^ to 0.0696 min^−1^. In addition, when the O_3_ aeration rate increased from 0.2 L min^−1^ to 0.4 L min^−1^, the values of *E*_EM_ and *E*_EO_ decreased to 188.21 kWh kg^−1^ and 2.57 kWh m^−3^ from 356.03 kWh kg^−1^ and 3.98 kWh m^−3^, respectively. Considering economy and practical application, 0.4 L^−1^ min was the optimal O_3_ aeration rate.

### 3.5. Possible Degradation Mechanism and Pathway of Metronidazole

#### 3.5.1. Active Species Evaluation

The quenching experiments of active species were performed using *p*-benzoquinone, isopropanol, and methanol as free radical trapping agents to explore the degradation mechanism of metronidazole in the 3D + O_3_ system. The results are displayed in Figure 9. The removal efficiencies of metronidazole were obviously reduced after the trapping agent was added. In the control experiment, the degradation rate reached 100% within 60 min, but it decreased to 90.9% and 91.4% after the addition of isopropanol and methanol. The treatment time was also delayed to 100 min before the complete degradation of metronidazole. Methanol and isopropanol are trapping agents for ·OH, and methanol with three α-H can react quickly with ·OH (k(·OH) = 9.7 ×10^8^ M^−1^ s^−1^) [53]. Therefore, ·OH played a certain role in the 3D electro-peroxone catalytic degradation of metronidazole. When *p*-benzoquinone was used as a probe, the degradation rates were significantly reduced, which were 80.7% and 96.4% at 60 min and 120 min, respectively. *P*-benzoquinone is the trapping agent of ·O_2_^−^. Thus, ·O_2_^−^ played a key role in metronidazole degradation. Shen et al. [54] also reported that ·O_2_^−^ was the reactive oxygen radical generated in Mg-doped ZnO catalytic ozonation. ·OH and ·O_2_^−^ were generally generated from the radical chain reactions of the decomposition of ozone and H_2_O_2_ that generated in situ on cathode as follows [47]:(8)$$O_{3}+H_{2}O+e^{-}\to \cdot OH+O_{2}+OH^{-}$$
(9)$$O_{3}+OH^{-}\to \cdot O_{2}{}^{-}+\cdot HO_{2}$$
(10)$$2\cdot HO_{2}{}^{-}\to O_{2}+H_{2}O_{2}.$$

#### 3.5.2. Possible Degradation Pathway of Metronidazole

The metronidazole was completely degraded within 60 min at optimum operating conditions. Meanwhile, the TOC removal rate was 43.2% in the first 20 min of reaction, and then increased to 50.5% within 60 min. Furthermore, a total of 12 intermediate products generated at reaction times of 10, 20, and 40 min in the 3D + O_3_ system were identified by LC-MS, and possible degradation pathways were proposed as illustrated in Figure 10. In pathway 1, the lateral N-ethanol group on metronidazole was directed and oxidized to product A and was then decarboxylated to product B (2-methyl-5-nitroimidozole), or A underwent hydroxylation of the nitro group to form D [3,55], and was further oxidized to E. In pathway 2, the nitro-hydroxylation product C of metronidazole was oxidized to D and then to F. In pathway 3, as previously reported [56], it began with a series of reduction reactions of the lateral nitro group to the nitroso product F, hydroxylamine product G and amino product H. After the N-ethanol group was oxidized to product I by active species, the C–N group on the imidazole ring was further oxidized and destroyed to form product J. In pathway 4, the N-denitration product K of metronidazole was generated [56,57], and then, the lateral methyl group was oxidized to carboxyl group to form product L. In conclusion, metronidazole was degraded into intermediates through hydroxyethyl cleavage, hydroxylation, nitro-reduction, N-denitrification, ring-opening, and other processes. It eventually oxidized and mineralized into low-toxic small molecular products, namely, CO_2_, H_2_O, and NO_3_^−^.

### 3.6. Toxicity Assessment

Acute toxicity LD_50_, bioconcentration factor, developmental toxicity, and mutagenicity of metronidazole and its degradation intermediates were predicted by Toxicity Estimation Software Tool (T.E.S.T Version 5.1.1) using the Consensus method and Quantitative Structure Activity Relationship methodologies [58,59]. The toxicity of some products cannot be predicted by T.E.S.T and, thus, they are not shown in Figure 11. The acute toxicity LD_50_ for rats of metronidazole was 3002.29 mg kg^−1^, which indicated “toxicity” (Figure 11a). Intermediate products C, F, and K were still regarded as having “toxicity” despite their enhancement in acute toxicity, and that of product H was weakened slightly. The bioconcentration factor of the final byproducts of all degradation pathways were significantly reduced (Figure 11b). Metronidazole and all of its intermediate products except J were developmentally nontoxic (Figure 11c). In particular, metronidazole was mutagenicity positive; except for the comparable mutagenic value of A and H and the higher mutagenic value of G, the mutagenic value of other products decreased (Figure 11d).

### 3.7. Stability and Reusability of Particle Electrode

The stability and reusability of the particle electrode are important indexes to evaluate its application prospect. Therefore, the recycling performance of the particle electrode for metronidazole elimination was tested under identical conditions. The cyclic electrocatalytic degradation of metronidazole was conducted with only refreshed 600 mL solution after each experiment. After four cycles of consecutive operations, as depicted in Figure 12a, no significant change was observed, and only a 5.4% decrease at 60 min and complete degradation was achieved after 80 min. The slight reduction in efficacy may be attributed to the slight loss of catalyst during the reuse of the particle electrode. The results showed that the particle electrode could work continuously without any activation treatment, which greatly reduced the operation cost. In the 3D + O_3_ system, organic substances adsorbed on particle electrodes could be electrochemically oxidized and in situ oxidized by O_3_, H_2_O_2_, and generated oxygen active species (·OH and ·O_2_^−^). Thus, the adsorption capacity of particle electrodes can be regenerated in situ, which increased the effective life of the particle electrode. Zhan et al. [24] reported that adsorption in the 3D + O_3_ process played an important role in the emission reduction of pollutants in the water environment. The recovered particle electrode was analyzed by XRD and compared with the original one, as displayed in Figure 12b. Compared with the XRD pattern of the original ZnO@BCFPs, the diffraction peaks of ZnO were still retained after use either once or four times. This finding confirmed the stability of ZnO on the particle electrode after the cyclic degradation process.

In general, the stability and reusability of the particle electrode were comparable to or even better than those of other reported materials [48,60]. In addition, commercial GAC is prone to slagging and pulverization, whereas ZnO@BCFPs remained intact after circle experiments. Thus, ZnO@BCFPs were promising particle electrodes.

## 4. Conclusions

A novel lightweight ZnO@BCFPs particle electrode was prepared and used in a 3D electro-peroxone process for metronidazole degradation. The optimal ZnCl_2_ dosage for particle electrode preparation was the mass ratio of ZnCl_2_ to eucalyptus sawdust of 3.2:1. The optimal condition for metronidazole degradation in the 3D electro-catalytic system was an Na_2_SO_4_ concentration of 0.05 mol L^−1^, a current density of 8.00 mA cm^−2^, a ZnO@BCFPs dosage of 10.00 g L^−1^, and a pH of 7.0, with a removal efficiency of 100% at 120 min. The degradation rate was enhanced greatly in the 3D electro-peroxone system after the addition of the particle electrode, and the total degradation time was shortened to 60 min. The quenching experiments of active species showed that ·OH and ·O_2_^−^ played an important role in metronidazole degradation. The acute toxicity LD_50_ and bioconcentration factor of intermediate products were decreased. Therefore, a fluidized ZnO@BCFPs particle electrode can avoid short circuit current and improve the efficiency of electrocatalysis, and the 3D electro-peroxone process with a fluidized particle electrode has a broad application prospect for the degradation and detoxification of organic pollutants in water.

## Figures and Tables

**Figure 1 materials-15-03731-f001:**
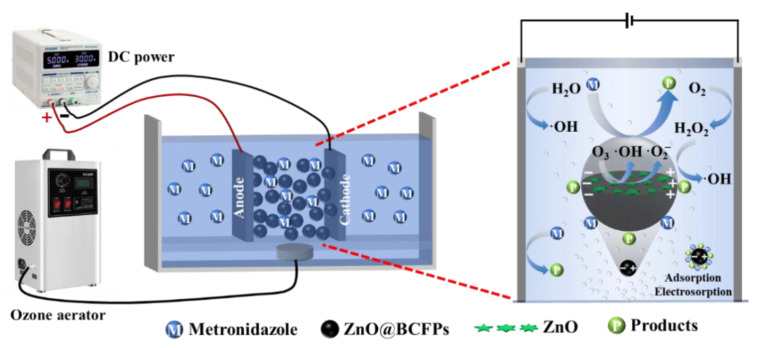
Schematic of self-made 3D electrocatalytic reactor.

**Figure 2 materials-15-03731-f002:**
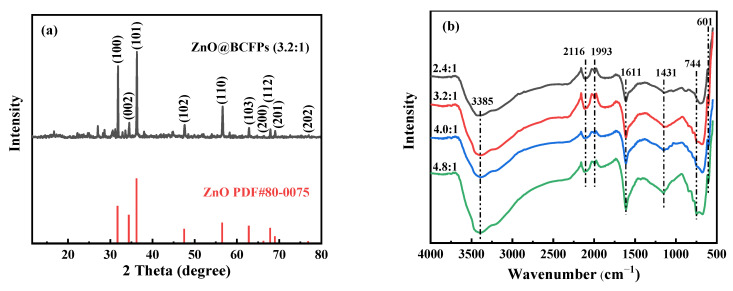
Characterization of particle electrode: (**a**) XRD, (**b**) FTIR and (**c**) EIS.

**Figure 3 materials-15-03731-f003:**
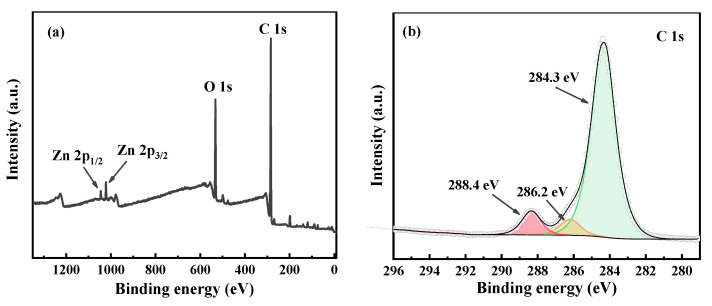
XPS spectra of particle electrode: (**a**) total survey, (**b**) C 1s, (**c**) O 1s and (**d**) Zn 2p.

**Figure 4 materials-15-03731-f004:**
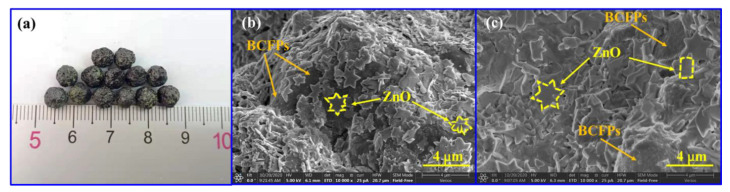
Optical picture (**a**) and SEM graph of particle electrode: (**b**) external surface and (**c**) internal surface (the dashed line symbols represent ZnO).

**Figure 5 materials-15-03731-f005:**
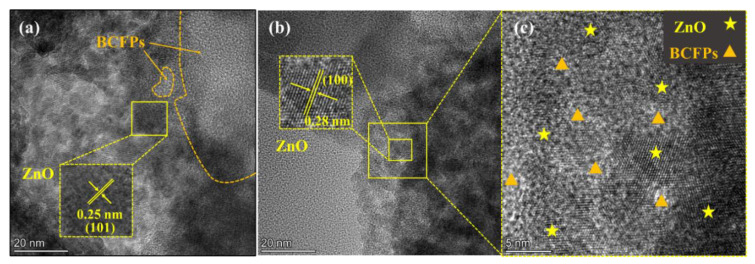
HRTEM images of particle electrode: (**a**−**c**) the lattice fringes and distribution of ZnO in the BCFPs.

**Figure 6 materials-15-03731-f006:**
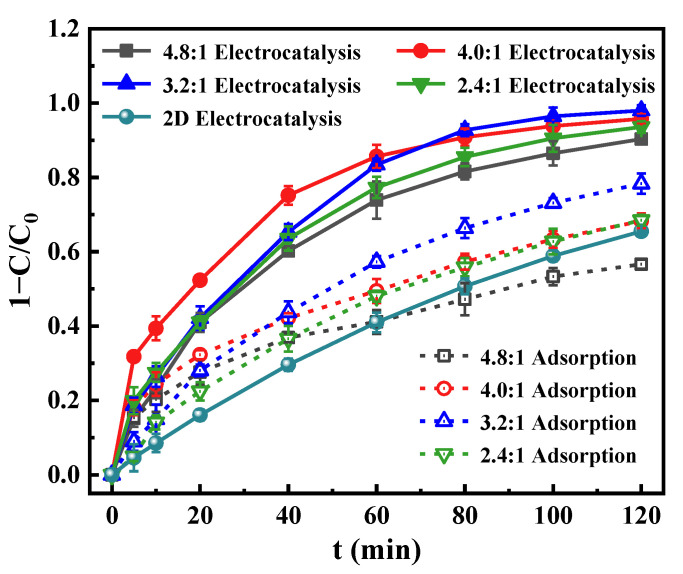
Effect of ZnCl_2_ dosage on the removal efficiency of metronidazole ([metronidazole]_0_ = 15 mg L^−1^, [Na_2_SO_4_]_0_ = 0.05 mol L^−1^, current density = 8.00 mA cm^−2^, particle electrode dosage = 6.67 g L^−1^, solution pH = 7, air aeration rate = 0.4 L min^−1^).

**Figure 7 materials-15-03731-f007:**
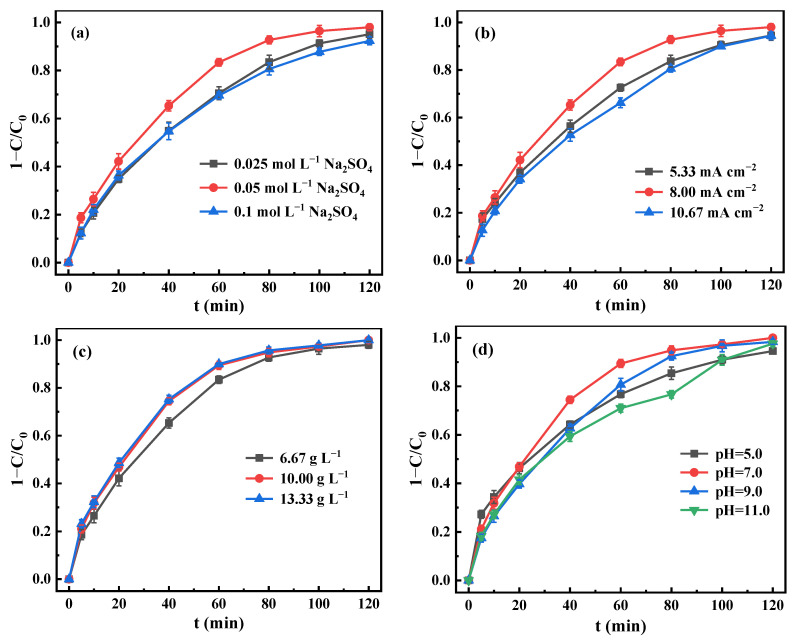
Effect of key parameters on removal efficiency of 15 mg L^−1^ metronidazole: (**a**) Na_2_SO_4_ concentration (current density = 8.00 mA cm^−2^, dosage = 6.67 g L^−1^, pH = 7.0, air aeration rate = 0.4 L min^−1^); (**b**) current density ([Na_2_SO_4_]_0_ = 0.05 mol L^−1^, dosage = 6.67 g L^−1^, pH = 7.0, air aeration rate = 0.4 L min^−1^); (**c**) particle electrode dosage ([Na_2_SO_4_]_0_ = 0.05 mol L^−1^, current density = 8.00 mA cm^−2^, pH = 7.0, air aeration rate = 0.4 L min^−1^); (**d**) solution initial pH ([Na_2_SO_4_]_0_ = 0.05 mol L^−1^, current density = 8.00 mA cm^−2^, dosage = 10.00 g L^−1^, air aeration rate = 0.4 L min^−1^).

**Figure 8 materials-15-03731-f008:**
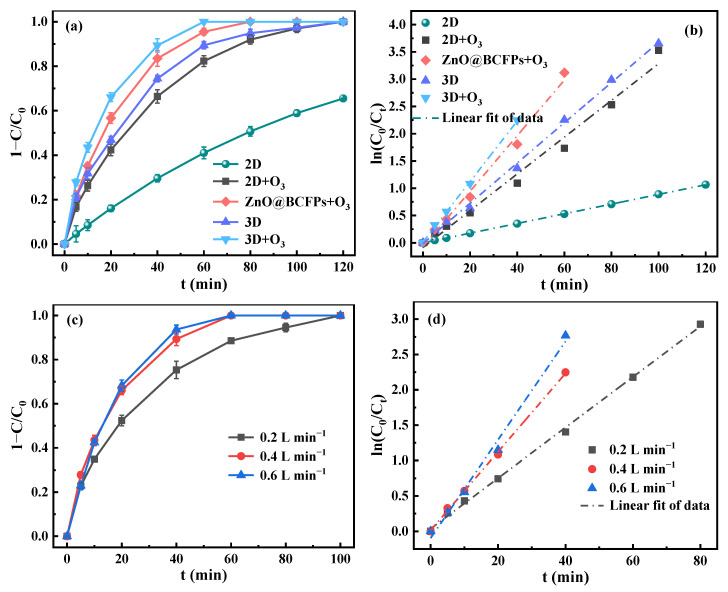
Comparison of removal efficiency in different systems (**a**) and (**b**); Effect of O_3_ aeration rate on removal efficiency (**c**) and (**d**). ([metronidazole]_0_ = 15 mg L^−1^, [Na_2_SO_4_]_0_ = 0.05 mol L^−1^, current density = 8.00 mA cm^−2^, dosage = 10.00 g L^−1^, pH = 7).

**Figure 9 materials-15-03731-f009:**
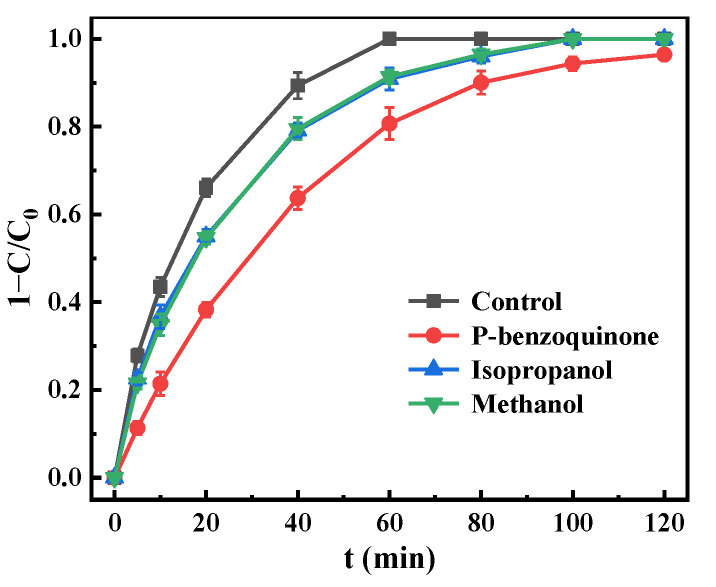
Effect of scavengers on removal efficiency. ([metronidazole]_0_ = 15 mg L^−1^, [Na_2_SO_4_]_0_ = 0.05 mol L^−1^, [scavenger] = 0.5 mmol L^−1^, current density = 8.00 mA cm^−2^, dosage = 10.00 g L^−1^, pH = 7, O_3_ aeration rate = 0.4 L min^−1^).

**Figure 10 materials-15-03731-f010:**
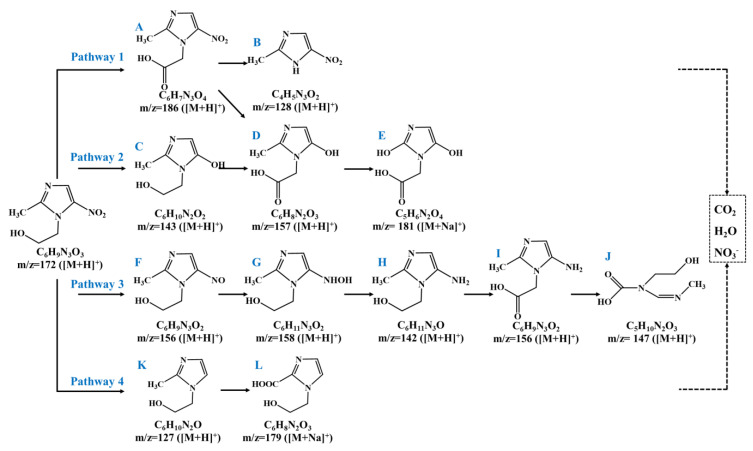
Possible degradation pathways of metronidazole: (**A**–**L**) intermediate products of metronidazole.

**Figure 11 materials-15-03731-f011:**
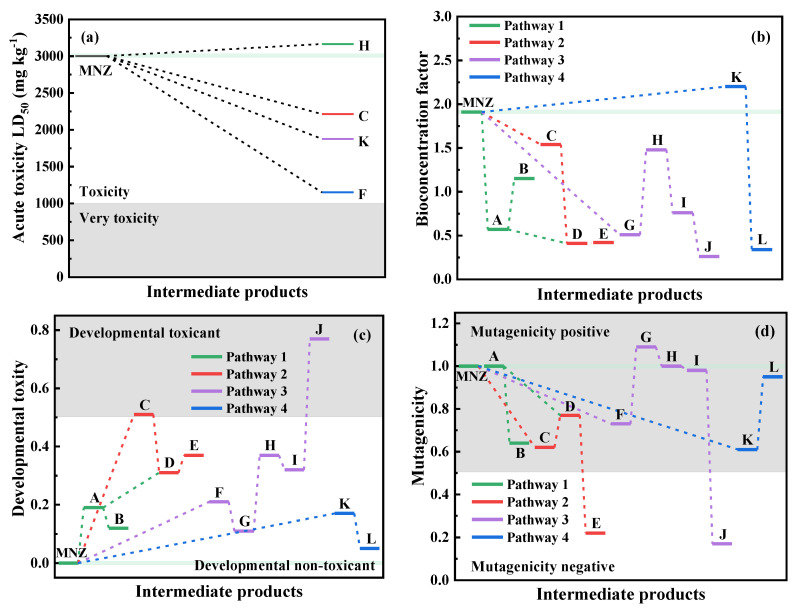
Toxicity assessment of metronidazole and its degradation intermediate products (metronidazole is abbreviated as MNZ): (**a**) acute toxicity LD_50_, (**b**) bioconcentration factor, (**c**) developmental toxicity and (**d**) mutagenicity. (**A**–**L**) intermediate products of metronidazole.

**Figure 12 materials-15-03731-f012:**
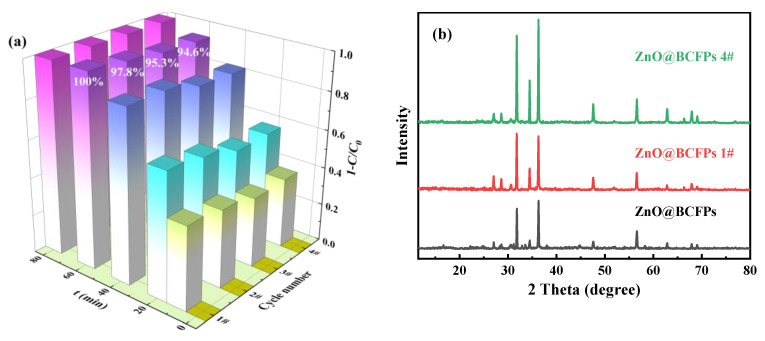
Reusability test of particle electrodes (**a**) and XRD of particle electrode before and after metronidazole degradation (**b**) ([metronidazole]_0_ = 15 mg L^−1^, [Na_2_SO_4_]_0_ = 0.05 mol L^−1^, current density = 8.00 mA cm^−2^, dosage = 10.00 g L^−1^, pH = 7, O_3_ aeration rate = 0.4 L min^−1^).

**Table 1 materials-15-03731-t001:** The fitted parameters of EIS data for particle electrodes with different mass ratio of ZnCl_2_ to eucalyptus sawdust.

Particle Electrode	R_0_ (Ω)	R_1_ (Ω)	R_2_ (Ω)
2.4:1	7.47	45.51	3.69 × 10^5^
3.2:1	6.81	45.40	2.05 × 10^5^
4.0:1	4.65	49.18	2.13 × 10^5^
4.8:1	5.53	47.01	1.45 × 10^6^

**Table 2 materials-15-03731-t002:** Fitting parameters of pseudo-first-order kinetic model and energy consumption in different systems.

Reaction Systems	Aeration Rate (L min^−1^)	*k*_obs_ (min^−1^)	R^2^	*E*_EM_ (kWh kg^−1^)	*E*_EO_ (kWh m^−3^)
2D	0.4	0.0089	1	820.66	16.01
2D + O_3_	0.4	0.0337	0.9836	452.24	4.22
ZnO@BCFPs + O_3_	0.4	0.0509	0.9886	-	-
3D	0.4	0.0369	0.9975	432.52	3.86
3D + O_3_	0.4	0.0555	0.9986	188.21	2.57
3D + O_3_	0.2	0.0358	0.9984	356.03	3.98
3D + O_3_	0.6	0.0696	0.9892	176.03	2.05

## Data Availability

The data presented in this study are available on request from the corresponding authors.

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
