# Peer review of "Fluidized ZnO@BCFPs Particle Electrodes for Efficient Degradation and Detoxification of Metronidazole in 3D Electro-Peroxone Process"

_materials, 2022, doi:10.3390/ma15103731_

Round 1

Reviewer 1 Report

This manuscript described about the synthesis of a new ZnO-embedded biomass carbon foam pellets (ZnO@BCFPs) using eucalyptus sawdust (waste), which was tested as fluidized particle electrodes in three-dimensional (3D) electro-peroxone systems for metronidazole degradation. I believe that this subject is appropriate for this Journal and it could be accepted for publishing after major revision of the manuscript.

Specific comments:

  1. The dosage of particulate electrode (ZnO@BCFPs) is given in g but must be expressed in gL-1. Please correct this during the whole manuscript.
  2. Considering the main disadvantage related to the energy consuming, authors should discuss this aspect by comparison with other tested systems (2D+O3) to determine the role of the electrode particulate.
  3. Authors determined metronidazole concentration by HPLC and identified the degradation sub-products. What about the mineralization process? At least for the optimum operating conditions, the analysis of TOC (total organic carbon) parameter to check/prove metronidazole mineralization should be done.

Reviewer 2 Report

The manuscript by Dan Yuan et al reports “Fluidized ZnO@BCFPs particle electrodes for efficient degradation and detoxification of metronidazole in 3D electro-peroxone process”. This article discusses the synthesized of self-shaped ZnO-embedded biomass carbon foam pellets (ZnO@BCFPs) material by coupling liquefaction, resinification, hydrothermal, and carbonization processes. The obtained materials were employed in three-dimensional (3D) electro-peroxone systems for metronidazole degradation as fluidized particle electrodes.

Overall, this work is interesting; however, some issues deserve clarification and should be carried out, which are not discussed detailed-wise. I recommend a major revision for this manuscript. Below are some issues and ideas for the author's consideration.

Comment 1: First of all, even though editing by EnPapers, the English language needs significant attention as some sentences are not quite correct and clear. You can find the grammatically corrected manuscript in the attached file.

Comment 2: The important research findings must be highlighted in the abstract. The general routine writing must be removed and should be highlighted with research findings.

Comment 3: The important research findings must be highlighted in the abstract. The general routine writing must be removed and should be highlighted with research findings.

Comment 4: The authors mention in the abstract that “Electrochemical advanced oxidation processes (EAOPs) have attracted extensive attention as effective technologies for the treatment of wastewater containing refractory organic pollutants because of their high efficiency, simple device structure, and environmental friendliness [9].”. How does this process compare to other processes in the literature? For example” photocatalysis”, a review demonstrated that Photocatalysis could remove antibiotics from wastewater effectively with different types of catalysts (10.1016/j.jwpe.2021.102089); authors can compare their process with the photocatalysis process explained in this review.

Comment 5: Provide the different methods available for treatments of those pollutants in the introduction such as flocculation, coagulation, photocatalysis, etc. I suggest an interesting work (10.1016/j.jclepro.2021.129934) that deals with the removal of an antibiotic using an efficient photocatalyst. The authors can compare the advantages of their process with the process of this study, especially in the by-products part (Possible degradation pathways).

Comment 6: there is a lack of characterization techniques such as XPS. Also, the morphology needs to be illustrated in a clear way as the given SEM is not significant. I suggest adding TEM images if it is possible (suspension mode).

Comment 7: What about the electrochemical characterization, I recommend adding it.

Comment 8: The authors mentioned that “It eventually oxidized and mineralized into low-toxic small molecular products, namely, CO2, H2O, and NO3−”, how did the authors demonstrate the mineralization? There is no analysis about the mineralization which is very important in the removal process, please perform the TOC analysis in order to confirm the mineralization of products? This needs to be fully addressed in the manuscript. Its true LCMS analysis can determine the by-products however total carbon organic can determine the mineralization.

Comment 9: authors didn’t work on industrial wastewater effluents, the authors used only synthetic aqueous solutions and they used metronidazole aqueous solution as simulated wastewaters. Please, add the results of the industrial wastewater effluents instead of synthetic aqueous solutions if it is possible, it will be more interesting.

Comment 10: How much practical is this process in real-life applications.

Author Response

Please see the attachment. In addition, the comment 2 and comment 3 are duplicate. Therefore,  we provide nine point-by-point responses to the reviewer’s comments.

Reviewer 3 Report

Reviewer Recommendation and Comments for manuscript materials-1683754 with the title: “Fluidized ZnO@BCFPs particle electrodes for efficient degradation and detoxification of metronidazole in 3D electro-peroxone process”, authors: D. Yuan, S. Wan, R. Liu, M. Wang, L. Sun.

The authors report the synthesis of a new material based on processes of liquefaction, resinification, hydrothermal and carbonization of ZnO / biomass. The newly obtained materials were used as electrodes in three-dimensional electro-peroxone (3D) systems for metronidazole degradation.

The article may be published after revision.

The main comments that I find useful for improving the quality of the article are presented below:

*General remark. Such studies have a very high relativity. The unknown component of the study is "eucalyptus sawdust". The composition of such a material varies depending on the year, humidity, drought, harvest period, altitude etc. Authors should enter as much data as possible on the composition and concentration of major components. Is there an optimal eucalyptus harvest period/growth stage?

*line 173. Figures 2a and b require more elaborate comments. What about the bands located at 2000/2250 cm-1?

*line 185. ”oxidation of anode”. Anode material is not electrochemical stable?

*Figure 4. Figure 4 needs to be commented on in more detail. In the absence of the catalyst, the variation in concentration over time is linear, while in the presence of the catalyst it is exponential. Without making a clear distinction between ab / adsorption and electrochemical degradation, does the global reaction kinetics seem to change?

*line 210. ”is to produce stronger active species”. What species?

*line 223. What it means ”repolarization”?

*line 227. ”current efficiency and polarization degree”. How has current efficiency been measured?

*line 337. The stoichiometry and Lewis structures of the species in reactions 6, 7, 9 and 10 must be checked.

* Figure 8 is unclear and needs to be replaced with a higher resolution.

*”Author Contributions” format is incorrect.

*The typos must be corrected.

ln/Ln

line 211. contaminates/ contaminantes

etc.

*The Materials journal require a specific format of references, authors must pay more attention in their writing. eg. no/no

Round 2

Reviewer 1 Report

The required changes were done by the authors. In my opinion, the paper can be accepted in current revised form. 

Reviewer 2 Report

The reviewers have done a great work in this revision. All comments have been addressed, improving this interesting manuscript. Therefore, I recommend accept it in the present form.